# Less-Invasive Approach to Early-Onset Scoliosis—Surgical Technique for Magnetically Controlled Growing Rod (MCGR) Based on Treatment of 2-Year-Old Child with Severe Scoliosis

**DOI:** 10.3390/children10030555

**Published:** 2023-03-15

**Authors:** Pawel Grabala, Ilkka J. Helenius, Kelly Chamberlin, Michael Galgano

**Affiliations:** 1Department of Pediatric Orthopedic Surgery and Traumatology, University Children’s Hospital, Medical University of Bialystok, Waszyngtona 17, 15-274 Bialystok, Poland; 2Paley European Institute, Al. Rzeczypospolitej 1, 02-972 Warsaw, Poland; 3Department of Orthopedics and Traumatology, Helsinki University Hospital, 00260 Helsinki, Finland; 4Department of Neurosurgery, University of North Carolina, Chapel Hill, NC 27516, USA

**Keywords:** early-onset scoliosis, EOS, magnetically controlled growing rod, MCGR

## Abstract

Background: Spinal deformities in children can be caused by various etiologies, such as congenital, syndromic, neuromuscular, or idiopathic. Early-onset scoliosis (EOS) is diagnosed before the age of ten years, and when the curvature continues to progress and exceeds a Cobb angle of 60–65 degrees, surgical treatment should be considered. Initial minimally invasive surgery and the implantation of magnetically controlled growing rods (MCGRs) allows for the noninvasive distraction of the spine, growing, and avoids multiple operations associated with the classic distractions of standard growing rods. Case presentation: A 2-year-old girl was admitted to our clinic with rapidly progressive thoracic scoliosis. The major curve of the thoracic spine Cobb angle was 122° at 30 months. No congenital deformities were detected. The surgical technique was the less-invasive percutaneous and subfascial implantation of MCGRs, without long incisions on the back and the non-invasive ambulatory lengthening of her spine over the next 4 years. Conclusions: MCGR is a safe procedure for EOS patients. It is extremely effective at correcting spinal deformity; controlling the growth and curvature of the spine as the child develops during growth; reducing the number of hospitalizations and anesthesia; and minimizing the physical and mental burden of young patients, parents, and their families.

## 1. Introduction

Spinal deformities in children and adolescents can be caused by various etiologies, such as congenital, syndromic, neuromuscular, or idiopathic [1]. According to the age of diagnosis, we can distinguish early-onset scoliosis (EOS), which is diagnosed before the age of ten years, and late-onset scoliosis, which is diagnosed under the age of ten years [2,3]. At the time of diagnosis, conservative treatment, rehabilitation, and a brace are recommended. In the absence of satisfactory results, when the curvature continues to progress and exceeds a Cobb angle of more than 60 degrees, surgical treatment should be considered [1]. The earlier surgery is started, the more complications we can expect during the entire course of treatment.

The most common procedure for EOS management is the surgical correction of the deformity with growing rods. The most significant disadvantage of this technique is the repetition of successive operations as the child grows, on average every 6 months under general anesthesia, which results in high rates of complications related to the procedure (up to 58%) [4]. MCGRs were designed in 2009 for the surgical treatment of spinal deformity in patients who are less than 10 years of age and have a thoracic spine height of less than 22 cm [5,6,7,8,9]. After the initial surgery and implantation of MCGRs, they allow noninvasive distractions, can reduce complications, and avoid multiple operations associated with the classic distractions of standard growing rods [6,10,11].

Since there are few descriptions and reports in the literature describing the point-by-point implantation techniques of MCGRs, we would like to share our many years of experience in using MCGRs. The aim of our work is to present the less-invasive surgical technique of the implantation of MCGRs and experiences of this procedure gained during the treatment of children with EOS less than 10 years of age, based on a case report of a young child in Europe with severe early-onset idiopathic scoliosis treated surgically with the use of MCGRs, followed by a 6-year observation period.

## 2. Case Presentation

A 2-year-old girl was admitted to our clinic with rapidly progressive thoracic scoliosis. Previously, the patient had undergone conservative treatment with a brace and physical therapy. The patient had no other disorders. The major curve of the thoracic spine Cobb angle was 70° at 16 months of age, 100° at 22 months, and 122° at 30 months (Figure 1).

Computed tomography and magnetic resonance imaging were performed. The spinal cord was intact. No other congenital deformities were detected. The patient was born through natural childbirth without any comorbidities (Figure 2).

All examinations, specialist consultations, and clinical statuses showed other causes of spinal deformity. We diagnosed early-onset idiopathic scoliosis with fast progression, which qualified for surgical treatment with growing rods (Figure 3). The flexibility was approximately 30%, so we performed anterior release and spine correction with intraoperative halo traction (Figure 4).

We placed the patient in a prone position on a Jackson table according to the standard procedure. Each procedure is carried out under the control of the neuromonitoring of the spinal cord. We selected the levels of stabilization of the upper and lower screws before the procedure, and after the analysis of the examined patient, we took X-ray images in the standing position and after assessing spine flexibility on the bending films. Then, after positioning the patient under the control of the C-arm, we marked the screw location levels on the skin, usually T2–T4 and L1–L3. Then, we made a skin incision, usually about 4–5 cm long (depending on the child’s height), to gain access to the levels of screw implantation only (Figure 5).

It is very important to prepare fixation points using two small separate incisions, and the rest of the spine should be left untouched. Next, we undertook the less-invasive implantation of screws, usually using a free-hand technique, but other techniques can also be used, such as navigation or robotic. After checking the adequate position of the screws with the C-arm and neuromonitoring, we made two subfascial tunnels on the left and right sides of the spine to insert the magnetic rods. Using a paean of the right size, we created rod tunnels immediately under the fascia not reaching the spine. To make it easier to guide the rods without damaging the muscles and to avoid perforation to the pleura, we inserted No. 16 drains into the created tunnels. First, we used a temporary rod to identify what the length of the final rod should be, as well as to assess the best method of obtaining the best rod bend (Figure 6a–c).

The next step is to properly cut and bend the magnetic rods. We always use one standard rod and one offset rod. After measuring and cutting the MCGRs, we need to test the efficiency of the rod distraction actuator. The testing of the actuator should be performed before bending the rods and after their proper modeling and contouring to exclude possible damage to the mechanism during the bending of the rod. A very important element of bending the rods is modeling the rods 1 cm above and below the ends of the actuator. This is to prevent mechanical damage to the actuator. However, this method is not perfect, and due to the inability to bend the actuator in any way, we must contend with a completely straight part of the rod with a length of 90 mm or 110 mm, depending on the type of rod used. Another important element is to plan the implantation of magnetic rods in such a way that the actuators of the rods are preferably at the same height. After the rods are properly prepared, we are able to proceed with their insertion. Our technique involves placing the first rod on the left side for right-sided curves and vice versa for left-sided ones. For this patient, we led the rod through the previously prepared tunnel, placing the end of the drain on the end of the rod (Figure 6a–c). We led the rod in the cephalad direction. After affixing the rod to the heads of the screws, we derotated it to an adequate sagittal position and temporarily blocked it. Subsequently, we repeated the process on the opposite side with the second rod. After rod insertion, we performed a gentle distraction distally across the base after the proximal set screws and finished final tightening. We do not routinely use a proximal cross-connector between the rods. For patients with poor bone quality, we use transverse process hooks at the top for the prevention of upper implant pull out. We then completed the final irrigation, the decortication of the bone at the screws, the covering of the bone grafts with vancomycin powder, and a standard closure. During surgery, we used safety distraction for the neuromonitoring of the spinal cord. There were no changes in the SSEP and MEP. No postoperative complications occurred. The patient was discharged on the sixth day after surgery with a brace (for three months). We started distraction after three months. We distracted the MCGRs with the external remote control (ERC) by 1.5–2.5 mm every 8–10 weeks. The radiographic parameters are listed in Table 1. Between 2017 and 2022, 28 distractions were performed (48 mm total).

No deep infections were observed during follow-up. At follow-up after 4.5 years, we observed a loss of distraction without other known radiological signs mentioned in the literature (Figure 7).

We decided to replace the 4.5 MCGR with a 5.5 MCGR. We performed revision surgery to replace the screws and MCGRs in January 2022 (Figure 8).

Intraoperatively, metallosis was observed around the screw–rod junction. After revision surgery, the patient was discharged on the fifth day after surgery without a brace. There was no external rod fracture, but there were many wear marks at the junction between the extended portion of the rod and actuator (Figure 9).

The removed rods did not function when checked with the ERC. After revision surgery, the child was examined and her new MCGRs were distracted and confirmed via X-ray.

## 3. Discussion

Initially, EOS can be treated conservatively with rehabilitation and a series of castings, corsets, or braces [1,2,12,13,14]. In the event of failure of this treatment and the further progression of the spinal curvature exceeding a Cobb angle of 60–65 degrees, surgical treatment should be considered [3,5,6,7,12,13,15,16,17,18].

After implantation, the MCGRs correct the curvature by initial spinal distraction; then, they maintain the resulting deformity correction by controlling both spinal curvature and spinal growth and promoting growth with further non-invasive instrumentation distractions. The non-invasive possibility of lengthening MCGRs reduces the risk of complications and significantly reduces the number of hospital stays, the amount of subsequent anesthesia (up to invasive spinal distraction), as well as the related mental and social feelings of children and parents [19,20,21].

The MCGR system might seem ideal, but it cannot be used for every patient with EOS, and the treatment with this system is limited. Not every patient can be implanted with this type of rod, due to their size and the limited possibilities of intraoperative bending. More precisely, they do not fit all curvatures and patients. There are also some risks of complications. Several studies have reported certain drawbacks and complications in the treatment course, such as deep wound infection, proximal junctional kyphosis (PJK), rod fracture, the failure of the rods to distract, and the auto-fusion of the spine before the end of treatment [20,22,23]. According to the data available in the literature, implant failure is most often associated with magnetic rod fracture or the loss of ability to continue distraction. In patients treated with the MCGR system in the medium and long-term observation period, the overall number of complications was found to be about 73%, but this is also related to the results of the treatment of congenital, neuromuscular, or syndromic scoliosis using this system. Furthermore, as we know, scoliosis with an etiology other than idiopathic has an increased risk of complications overall [9,11,22,23,24,25]. Other studies have also reported several unique complications, including rod slippage, mismatches between the target and achieved distraction length, metallosis, and actuator pin fracture. In comparison to other surgical techniques for EOS treatment, MCGRs can be inserted as a less-invasive surgical technique with a mini-open approach [26,27,28]. In order to limit the high risk of spontaneous spinal fusion in a growing spine [29,30,31] during surgery procedures with a wide approach to the spine, MCGRs should be placed subfascially. Metallosis resulting from the use of MCGRs appears to be of significant clinical importance (Figure 9). Although there are no studies confirming the toxic effects of this metallosis on soft tissues and the body, MCGR instruments should be removed at the end of the growth-promoting procedure, and final spine instrumentation should be made [27,29]. The use of MCGRs is still a safe and effective surgical technique in patients undergoing primary EOS surgery [5,6,7,10,17,18,22,23,32,33,34,35,36].

## 4. Conclusions

MCGR is a safe surgical technique in patients undergoing primary EOS surgery and is currently widely used. The MCGR system can be used as a less-invasive procedure that allows for the avoidance of many periodic invasive procedures for children, reducing the number of elective hospitalizations and anesthesia to a minimum and lessening the physical and mental burden of young patients, parents, and their families. Moreover, lengthening on an outpatient basis represents an advancement in the treatment of EOS.

## Figures and Tables

**Figure 1 children-10-00555-f001:**
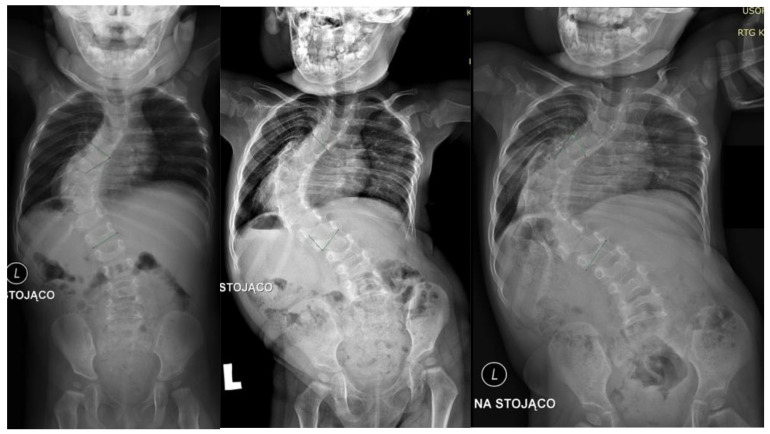
Showed standing: (from left to right) AP X-rays at 16 months old (70 degrees), 22 months old (100 degrees), and 30 months old (122 degrees).

**Figure 2 children-10-00555-f002:**
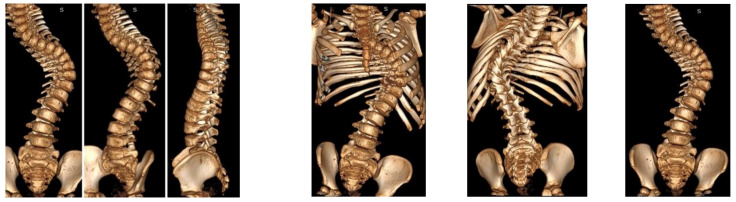
CT-3D preoperative images.

**Figure 3 children-10-00555-f003:**
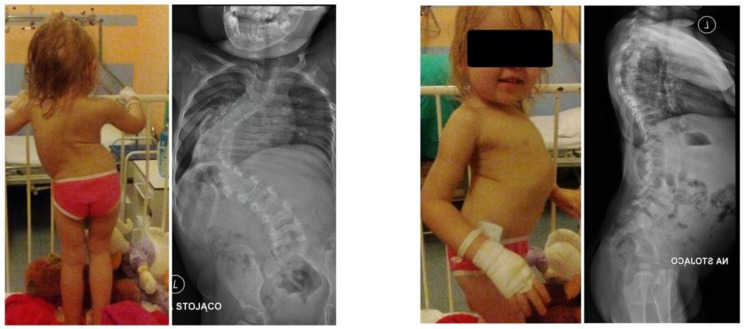
X-rays and clinical preoperative images.

**Figure 4 children-10-00555-f004:**
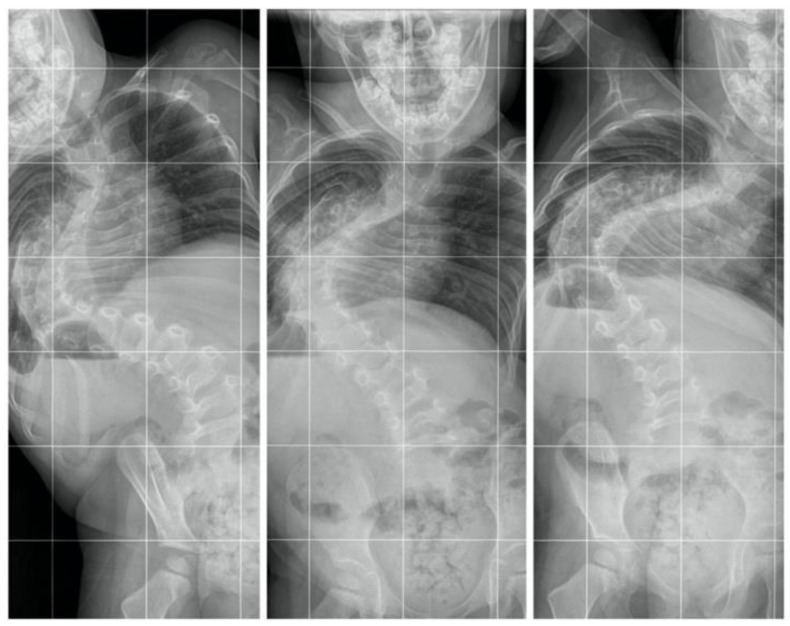
AP standing bending films.

**Figure 5 children-10-00555-f005:**
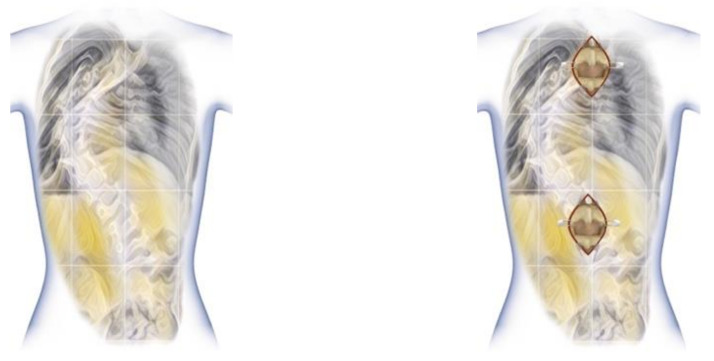
Less-invasive approach for upper and lower spine.

**Figure 6 children-10-00555-f006:**
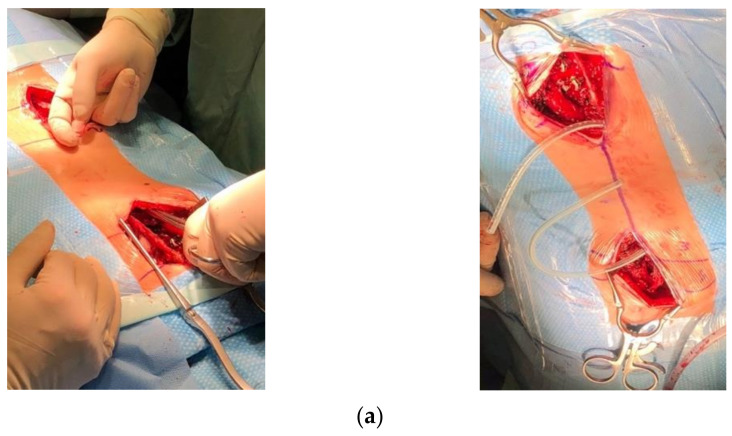
(**a**) Intraoperative pictures: using a paean, we created rod tunnels immediately under the fascia not reaching the spine. (**b**) Using drains as a guide for the insertion of temporary MCGRs. (**c**) Inserting MCGRs followed by drains.

**Figure 7 children-10-00555-f007:**
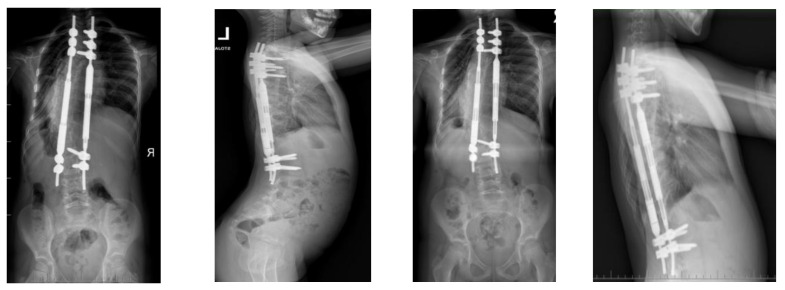
X-rays at follow-up after 1 year (**left** AP and LAT) and at follow-up after 4.5 years (**right** AP and LAT) before revision surgery for MCGR replacement.

**Figure 8 children-10-00555-f008:**
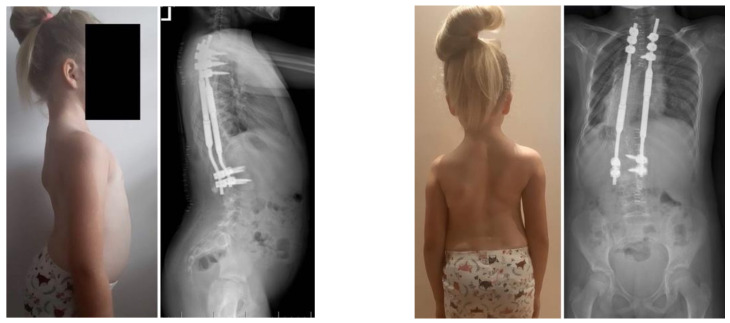
X-rays at 5-year follow-up, after revision surgery for MCGR replacement.

**Figure 9 children-10-00555-f009:**
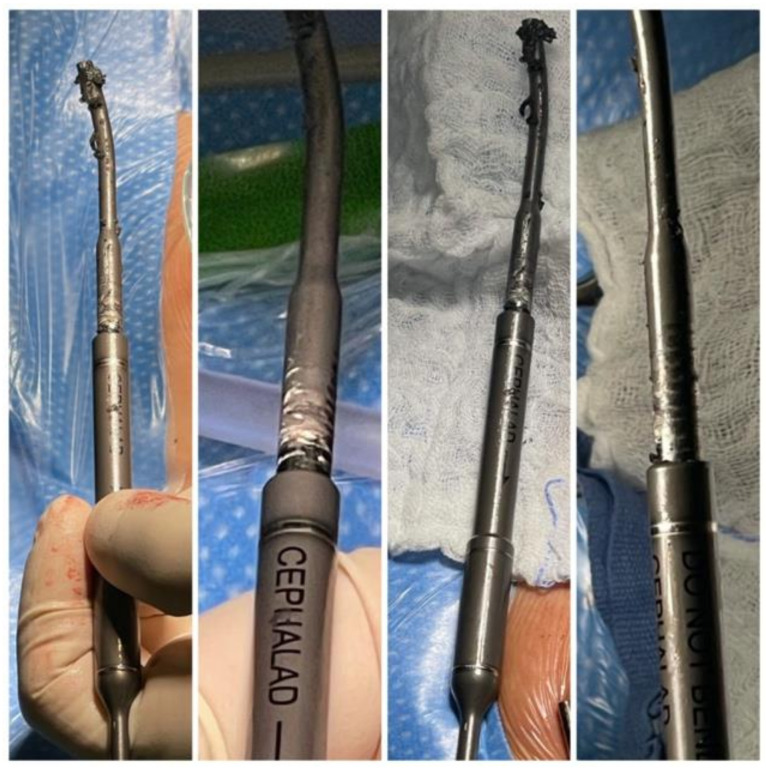
MCGRs and metallosis during revision surgery.

**Table 1 children-10-00555-t001:** Radiological parameters before and after surgical treatment.

	Pre-Operative (February 2017)	After Anterior Release and MCGR Placement	At 1-Year Follow-Up	Just Before Revision (December 2021)	At 5-Year Follow-Up (after Revision and MCGR Replacement) (February 2022)
T1-T12 (cm)	10.40	12.6	13.6	17.0	18.2
T1-S1 (cm)	18.90	21.4	22.7	28.5	29.8
Main Curve Cobb Angle (degrees)	120	51	46	42	40
Thoracic Kyphosis Cobb Angle (degrees)	41	36	28	22	34
Age (Follow-up)	2 5/12	2 6/12	3 6/12 (12 months)	7 3/12 (58 months)	7 5/12 (60 months)

## Data Availability

Not applicable.

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
