# Peer review of "Less-Invasive Approach to Early-Onset Scoliosis—Surgical Technique for Magnetically Controlled Growing Rod (MCGR) Based on Treatment of 2-Year-Old Child with Severe Scoliosis"

_children, 2023, doi:10.3390/children10030555_

Round 1

Reviewer 1 Report

The authors describe a clinical case of early onset scoliosis in children with Magnetically Controlled Growing Rod (MCGR) treatment.

However, this article has several weaknesses that make it incompatible for publication in this journal.

The authors describe an isolated clinical case with 5 years of follow-up. There are series of more than 50 patients with 10 years of follow-up that reflect much better the accumulated experience in this type of treatment, so the present article adds little new to the available literature.

This technique cannot really be considered minimally invasive, since as the authors themselves describe, the patient received 2 incisions of 5 cm in each area. I agree that it is not a continuous incision along the entire length of the spine, but it cannot be considered minimally invasive. Vertebral body tethering by thoracoscopy, on the other hand, could be included in this classification, as it involves minimal incisions (5-10 mm) through which screws are fixed in the hemithorax with greater convexity that have a kind of rail through which a guide is subsequently introduced that tightens the curve with the patient's own growth, without the need for reinterventions or retensioning.

On the other hand, figures with several images should be presented with letters (Figure 1A, 1B...) that should be described in the figure legends to make them easier to understand for the readers of the article. If you put several images in a row in the same figure and do not explain them, you lose the sense of the information you want to convey.

I recommend authors to reformulate the structure of the article and orient it as a systematic review or meta-analysis of the results of this technique.

Author Response

Dear Sir,

Thank you for your valuable review and comments.

I totally agree with you that this is not a fully minimally invasive surgical technique, but on the other hand, it is minimally invasive, because most surgeons operating on children with spinal deformities children less then 10 years old make a wide cut through the entire spine, thus exposing them to catastrophic effects and disability - the occurrence of spontaneous spinal fusion and inhibition of its growth.

The child will grow and the spine will fail, there will be respiratory failure and related complications.

Thoracoscopic techniques are not applicable to children less than 10 years old, except for the spinal mobilization technique known as anterior release.

Gold standard and surgical technique for early onset scoliosis are standard growing rods, wide approach and repeating the surgery every 6 months for mechanical distraction of spine under anesthesia. Our technique is really non-invasive like other techniques, but not minimally invasive like percutaneus screw placement. 

However, I agree that the text needed to be reformatted and changed, so that's exactly what I did. I have also changed the title for: „Less-invasive”.

Thank you very much for your review.

Kind regards,

Pawel Grabala

Reviewer 2 Report

1. Comment on how thoracic spine height is measured in a child. If 22 cm is the target how can accuracy of measurement be determined with a large curve?

2. 6 year monitoring brings patients age to pre-puberty. Postulate what might occur during puberty and how you would monitor and treat.

3. Was Respiratory and cardiac function normal prior to surgery a dis it normal now? Any precautions taken?

4.What advantage was doing both CT and MRI ? Would MRI alone reduce costs  and yield similar results?

5.Can the procedure be performed with a posterior technique or must it only be performed anterior?

6.Is bracing or traction of any advantage pre or post op in delaying the progression of the cure. If bracing is considered then what type of brace?

6. What is OR and equipment cost?

7.Are there any serious complications documented with this procedure?

8.Why loss of distraction at 4.5 years? Is this typical and must surgery be performed or can it be delayed?

9.Should fusion extend to iliac crest? If so why?

Author Response

Dear Sir,

Thank you for your valuable review and comments.

  1. Comment on how thoracic spine height is measured in a child. If 22 cm is the target how can accuracy of measurement be determined with a large curve? T1- T12 height: vertical distance (cm) measured between the middle of the superior end plate of T1 and the middle of the inferior end plate of T12 - Michael N, Carry P, Erickson M, et al. Spine and thoracic height measurements have excellent interrater and intrarater reliability in patients with early onset scoliosis. Spine (Phila Pa 1976) 2018;43:270e4.
  2. 6 year monitoring brings patients age to pre-puberty. Postulate what might occur during puberty and how you would monitor and treat.

As I mentioned, surgical treatment in children less than 10 years of age is completely different from that in children more than 10 years of age. When a child treated with MCGR reaches proper skeletal maturity, in most cases the implants are removed and posterior spinal fusion is performed. Spontaneous fusion occurs in a small number of patients.

  1. Was Respiratory and cardiac function normal prior to surgery a dis it normal now? Any precautions taken?

It is impossible to perform spirometry in a 2-year-old child, but there were no other problems, as described in the presentation of the patient's treatment.

4.What advantage was doing both CT and MRI ? Would MRI alone reduce costs  and yield similar results?

For me as an operator - there was no benefit, but the child came to our clinic with a computer tomography done. I always order an MRI, that is enough for me. I do computed tomography only for congenital scoliosis with a bone defect (but not always), and for patients who underwent spinal fusion with implants and need revision surgery. All I need for scoliosis surgery is an X-ray and an MRI.

5.Can the procedure be performed with a posterior technique or must it only be performed anterior?

For children less than 10 years old  the best technique and the most less-invasive technique with least risk of complications

6.Is bracing or traction of any advantage pre or post op in delaying the progression of the cure. If bracing is considered then what type of brace?

Surgical treatment should be postponed as far as possible. If positive treatment, such as rehabilitation and a corset, or brace works and delays the progression of the curvature, it should be done as long as possible. But if the curvature progresses despite all methods and exceeds 60-70 degrees, the progression of the curvature should be stopped surgically, because the child's lungs will not develop due to progressive scoliosis and it will affect the whole future life. There are many different types of corsets with proven effectiveness that we can use.

  1. What is OR and equipment

The cost of treatment is high. The cost of the implants alone is over 30,000. EUR. But considering that after treatment with the MCGRs technique, the child does not need to perform another operation every 6 months and repeat it up to 10, sometimes 20 times, the total cost of treatment is much less than with another technique.

7.Are there any serious complications documented with this procedure?

It all depends on the operating center, the surgeon and his experience. Complications are described in the literature up to 70%. In our group of patients (approx. 50) with a 5-year observation period, the worst complication was pull out of the upper screws (1 patient), broken rod (3 patients), deep infection (1 patient), pull out of the lower screws (1 patient). These complications did not affect the detriment of health. There were no neurological complications. We intend to publish our study soon. However, we are aware of cases of spinal fracture during MCGR treatment.

8.Why loss of distraction at 4.5 years? Is this typical and must surgery be performed or can it be delayed?

There are many reports in the literature about the cessation of the MCGR elongation mechanism after 2 years from implantation. With each subsequent year, the risk of losing the possibility of extension increases.

The producer's recommendations are to replace the rods with new ones after 2 years if we find a problem with the distraction mechanism.

9.Should fusion extend to iliac crest? If so why?

Extending the stabilization to the sacrum and pelvis is recommended only for patients with spinal deformities with neurogenic etiology like CP.

Thank you very much for your valuable review.

Kind regards,

Pawel Grabala

Reviewer 3 Report

There are some grammatical and typographic errors in the manuscript which need to be rectified. 

  Introduction  Very long. Can focus on MCGR in severe deformities only What is the purpose of this report?   Case report Although the study is described as a surgical technique,  the surgical procedure is not very clearly presented here in the case report. A greater detailed procedure is presented later. The current manuscript on the other hand is too long for a case report.    Discussion Is more in the form of a text book chapter. Is it possible to confine the entire discussion based on what is planned to be more relevantly studied?  Images can be more clearer In the current form, the discussion is very long. Pls modify Limitations need to be added   Conclusion is very long. Needs to be shorter       

Author Response

Dear Sir,

Thank you very much for your review and your valuable comments. At the outset, I would like to point out that we wanted to describe the most minimally invasive treatment of scoliosis in children less than 10 years old during the growth, which we use ourselves. It should be emphasized that the treatment of spinal deformities in children and adolescents differs significantly from the age of the child at which we treat such a child.  In most cases, in children more than 10 years of age, with bone maturity reached, or during puberty and after puberty, the gold standard for surgical treatment is posterior spinal fusion with wide open approach from T2 to L4, depends on type the main curve, or new techniques - VBT procedures. These technique cannot be usued cannot be used in children less than 10 years old. For children less than 10 years old  we cannot surgically interfere with the spine as in children over 10 years of age, because this spine will stop growing and the child will be disabled. The gold standard is limited fusion (only 4-5 levels) for congenital scoliosis and standard growing rods. The standard growing rods are placed on screws inserted on 4 level - usually 2 thoracic and 2 lumbar vertebrae. Standard growing rods are mostly implanted via long incision from T2-T4 to L1-L3 in the most children's hospitals. After this long approach, and screws and rods placement, the child is operated every 6 months, new incision and mechanical distraction in general anesthesia. The child can have 12-20 new surgeries until bone maturity and final spinal correction surgery with fusion. There is no other treatment. All other implants for children with the spinal deformity less than 10 years old like VEPTR, Shilla are even more invasive. So our technique of mini-invasive insertion of magnetically controlled rods through two small incisions, not opening the entire spine of a child, really saves the tissue and the baby. The little patient quickly recovers and becomes active. We do not perform further operations every 6 months to lengthen the  spine as the child grows to maturity. Distractions are performed without surgery, painlessly, ambulatory. This really improves the quality of life of the youngest patients, even in such severe and extremely complicated scoliosis, the treatment of which is described by us. We wanted to show and make people aware that even difficult cases can be treated better, giving hope for a normal life. It may not really be a mini-invasive technique, but it is less invasive.

The text has been significantly shortened. Only the most important things remain. The surgical technique was transferred from the discussion to the presentation of the treatment. The discussion has been shortened to the most important points. Limitations have been added and conclusions have also been shortened. The title has been changed to "less-invasive approach". The entire article was edited by the professional English language editor at the MDPI office (certificate attached).

I hope the new version is much better and clear.

Thank you very much for your review.

Kind regards,

Pawel Grabala

Round 2

Reviewer 1 Report

Accept in present form

Reviewer 3 Report

The recommended changes have been added. The manuscript can be accepted in the current form